# Two Cases of Autoimmune Thyroid Disorders after COVID Vaccination in Dialysis Patients

**DOI:** 10.3390/ijms231911492

**Published:** 2022-09-29

**Authors:** Georgios Lioulios, Ioannis Tsouchnikas, Chrysostomos Dimitriadis, Panagiotis Giamalis, Eva Pella, Michalis Christodoulou, Maria Stangou, Aikaterini Papagianni

**Affiliations:** Department of Nephrology, School of Medicine, Aristotle University of Thessaloniki, General Hospital “Hippokratio”, 54642 Thessaloniki, Greece

**Keywords:** ESRD, hemodialysis, peritoneal dialysis, SARS-CoV-2, vaccination, BNT162b2, Grave’s disease, Hashimoto’s disease

## Abstract

SARS-CoV-2 infection and vaccination have been associated with autoimmune thyroid dysfunctions. Autoimmune/inflammatory syndrome induced by adjuvants (ASIA) and molecular mimicry have been referred to as potential causes. Such a case has not been reported in immunocompromised end-stage renal disease (ESRD) patients. Herein we present two dialysis patients with no previous history of thyroid disease who developed immune mediated thyroid disorders after BNT162b mRNA vaccine against SARS-CoV-2. The first patient is a 29-year-old man on hemodialysis diagnosed with Grave’s disease four months post-vaccination and the second one is a 67-year-old female on peritoneal dialysis who developed Hashimoto’s thyroiditis two months post-vaccination. Grave’s disease is uncommon in dialysis patients, whereas Hashimoto’s thyroiditis has a higher incidence in this population. Time proximity in both cases suggests potential causality. To our knowledge, this is the first report of de novo immune-mediated thyroid disorders in dialysis patients following vaccination against SARS-CoV-2.

## 1. Introduction

SARS-CoV-2 causing COVID-19 disease emerged during the last few years and proved to be a pathogen not only causing respiratory infection but also affecting multiple organs, including the thyroid, resulting in deleterious yet unpredictable complications [1]. Vaccination against SARS-CoV-2 provides immune protection and represents the world’s main defend mechanism, able to protect from infection or even ameliorate its clinical symptoms. Two of the most used vaccines are based on novel mRNA technology [2]. COVID-19 disease per se and, to a much lower degree, the new technology vaccines against SARS-CoV-2 have been associated to various thyroid disturbances in the general population, emerging as soon as three days post-vaccination or rather late, with the delayed incidents occurring a few months following last dose administration [3,4,5,6,7,8,9,10,11].

Thyroid dysfunction prevalence has been reported to be higher in end-stage renal disease (ESRD) patients than in the general population [12]. While hypothyroidism is rather common, Grave’s disease is a rare clinical entity in the dialysis setting. Very few well-demonstrated cases of Grave’s disease have been described in the literature so far [13,14,15].

In this paper we describe the onset of autoimmune thyroid dysfunctions during the post-vaccination period in two ESRD patients undergoing dialysis.

## 2. Cases Presentation

### 2.1. Case 1

A 29-year-old man on a thrice weekly chronic maintenance hemodialysis (HD) program since the age of six due to chronic pyelonephritis presented for evaluation, reporting frequently relapsing palpitations and mild heat intolerance during the last month. He was afebrile, heart beats were 88/min and other clinical examination was unremarkable. Hemodialysis was performed through a central venous catheter for the last six years. Previous medical history was unremarkable, except from a renal transplantation twenty years ago which failed six months later. His current medication consisted of erythropoietin, iron, calcium supplements as phosphate binders and antihypertensives, including atenolol. Laboratory workout revealed a substantial drop in serum thyroid stimulating hormone (TSH) levels to 0.008 mIU/L (normal range (NR): 0.38–5.33 mIU/L), with raised serum free thyroxine (fT4) (1.9 ng/dL; NR: 0.6–1.49) and free triiodothyronine (fT3) (5.07 pg/mL; NR: 1.7–4.6 pg/mL) levels. TSH values measured twice a year for the last fifteen years during routine follow-up had been consistently within normal limits suggestive of normal thyroid function. He had been vaccinated with two doses of BNT162b mRNA vaccine against SARS-CoV-2, with the last dose administered 120 days before his presentation. Thyroid ultrasonographic findings included heterogeneous echotexture with diffuse hypervascularity and without nodules, while thyroid scan with Tc99 revealed a diffuse increased uptake, slightly more intense by the inferior left lobe (Figure 1). He tested positive for antibodies against TSH receptor (TSI) (2.9 IU/L; NR: <1.75 IU/L), while testing for antibodies against thyroglobulin (TG) and thyroid peroxidase (TPO) was negative. The patient was diagnosed with Grave’s disease and thiamazole was prescribed at an initial dose of 5 mg per day. He became asymptomatic three days later. The thiamazole dose was increased to 10 mg after the first week, and as thyroid laboratory values did not significantly improve after one month, it was further escalated to 20 mg per day. Six months after treatment initiation, the patient was still asymptomatic, serum TSH levels (0.025 mIU/L) were still low and fT3 and fT4 levels returned to normal (4.1 pg/mL and 1 ng/dL, respectively). At this time point, the patient was infected by SARS-CoV-2, showing only mild clinical symptoms, mainly low-grade fever and malaise, without any symptoms of respiratory infection. However, two months later, during laboratory reevaluation, a new deterioration of thyroid hormone levels was observed, with increased fT3 and fT4 levels and a further reduction in TSH levels. The thyroid hormone fluctuation is shown in Figure 1. After endocrinology consultation, it was decided to apply a wait-and-see policy since the patient remained asymptomatic. Then, two months after thyroid dysfunction exacerbation, TSH levels showed an upwards tendency, with fT3 and fT4 levels within the normal range.

### 2.2. Case 2

A 67-year-old woman, treated with continuous cyclic peritoneal dialysis (CCPD), presented complaining of generalized weakness, fatigue, mood disorder and hoarseness. The patient was on peritoneal dialysis (PD) due to ESRD for seven years, and her primary renal disease was amyloid light-chain (AL) amyloidosis, which had been successfully treated by autologous bone marrow transplantation. She was prescribed antihypertensives and had no history of thyroid disease, with normal TSH values during her six-monthly laboratory evaluation during her PD course. She was afebrile, her blood pressure was 102/62 mmHg and heart rate was 72/min. The present laboratory evaluation revealed mild hyponatremia (131 mEq/L), increased creatine phosphokinase levels, exceptionally high TSH serum levels up to >495 mIU/L (overflow) and a concomitant reduction in fT4 (0.21 ng/dL) and fT3 (1.65 pg/mL) levels. Antibodies against TG and TPO tested positive (>5000 IU/mL and 1504 IU/mL, respectively). Ultrasonography revealed an enlarged hypoechoic thyroid gland, with a few echogenic septations and hypervascularity (Figure 2). She was diagnosed with Hashimoto’s thyroiditis and rhabdomyolysis. She had been vaccinated against SARS-CoV-2 with two doses of BNT162b mRNA vaccine fifty days before the diagnosis of hypothyroidism. Levothyroxine sodium was prescribed. The patient became asymptomatic five days later, and two months afterwards during her routine re-evaluation, she was still asymptomatic and thyroid hormone levels had returned to reference range. 

## 3. Discussion

In this paper, we present two cases of new onset autoimmune thyroid disorders diagnosed during the post-vaccination period against SARS-CoV-2 in two ESRD patients in our center: a 29-year-old man undergoing HD with new onset of Grave’s disease, who showed improvement in thyroid function with treatment and relapse after SARS-CoV-2 infection, and a 67-year-old woman treated by CCPD, with new onset of Hashimoto’s thyroiditis.

Grave’s disease is rare in patients undergoing chronic HD, with only eleven cases having been reported in the literature from 1988 to 2003 [15]. Hypothyroidism, on the other hand, is much more common in ESRD patients than in the general population, especially as a subclinical disease, and has been recently recognized as a substantial risk factor for cardiovascular disease [16,17]. Nonetheless, although thyroid-related autoantibodies have been increased in this population, only a small portion of hypothyroidism can be attributed to autoimmunity [12].

Autoimmune thyroid diseases, Grave’s disease and Hashimoto’s thyroiditis have been previously associated with both infection from SARS-CoV-2 and vaccination against. New onset and aggravation of hyper- and hypothyroidism have been reported following infection and vaccination [3,4,5,8,9]. The underlying cause has not been clearly demonstrated. Among mechanisms proposed to lead to thyroid autoimmunity following vaccination is the autoimmune/autoinflammatory syndrome induced by adjuvants (ASIA), [9], which was introduced by Shoenfeld in 2011 [18]. Molecular mimicry between the spike protein encoded by mRNA in BNT162b has been proposed as a more convincing idea, as several studies have shown common epitopes between the thyroid gland and SARS-CoV-2 proteins [19]. Apart from molecular similarities, an additional mechanism proposed for thyroid disease after infection with SARS-CoV-2 is the inflammatory cascade caused by the virus per se. Increased circulating interleukin-6 (IL-6), which has been associated with Grave’s disease, and increases in IL-1 receptor antagonist (L1RA), C-C motif chemokine ligand 2 (CCL2), CCL8, C-X-C motif chemokine ligand 9 (CXCL9) and CXCL16 are reported to lead to thyroid autoimmunity [8,20,21]. Of note, in addition to the above proposed mechanisms, immune dysfunction and relative immunosuppression in ESRD must be taken into consideration as a potential cause. Alterations of adaptive immunity are prominent in these patients and resemble natural aging, albeit with distinct discrepancies present. Severe lymphopenia with a reduction mainly in naïve T and B cell subpopulations results in a less intense immune response and consequently delayed pathogen clearance [22]. Characteristically, the median time until clearance of SARS-CoV-2, even after clinical remission, was rather elongated in dialysis patients in comparison to patients without kidney disease [23]. Moreover, antibody production appears to be impaired in ESRD patients [24] potentially with lower affinity to the corresponding antigen and broader antigen recognition capacity. Finally, regulatory T cells are also diminished in the setting of ESRD, leading to inappropriate elongation of the anyway altered response [25]. For all these reasons, both natural infection and vaccination elicit a weak yet extended immune response, with a tendency to cross-reactivity to non-pathogenic antigens, thus leading to autoimmunity.

The onset of autoimmune thyroid disease was observed relatively long after the second vaccination dose in our cases, especially regarding the period of four months for the diagnosis of Grave’s disease. Noteworthily, the symptoms of this first case had appeared about 30 days before the diagnosis, suggesting that the disease onset was three months after the second vaccination dose. The current literature reports that symptomatology may be initiated 2–60 days post-vaccination in otherwise healthy individuals [8,9]; however, this time period may be expanded up to 90 days when thyroid disease follows COVID-19 infection [4]. The presence of ESRD may have an impact on the longer interval between vaccination and disease manifestation.

To our knowledge, this is the first report of autoimmune thyroid diseases diagnosed shortly after vaccination against SARS-CoV-2 in patients treated by dialysis. No case of such overt Hashimoto’s thyroiditis has been reported in PD patients following vaccination. Impressively, the rarity of hyperthyroidism in hemodialysis patients, the development of Grave’s disease post-vaccination against COVID-19 and subsequent relapse after infection with SARS-CoV-2 reinforces the causative relationship between the two events. Nonetheless, despite the rarity of immune-mediated thyroid disorders, and especially Grave’s disease, in dialysis patients, none of the above proposed mechanisms have been proven to associate vaccination against SARS-CoV-2 with such disturbances. Moreover, the presentation of the above cases does not aim to discourage vaccine administration, but to provide awareness and encourage further investigation.

## Figures and Tables

**Figure 1 ijms-23-11492-f001:**
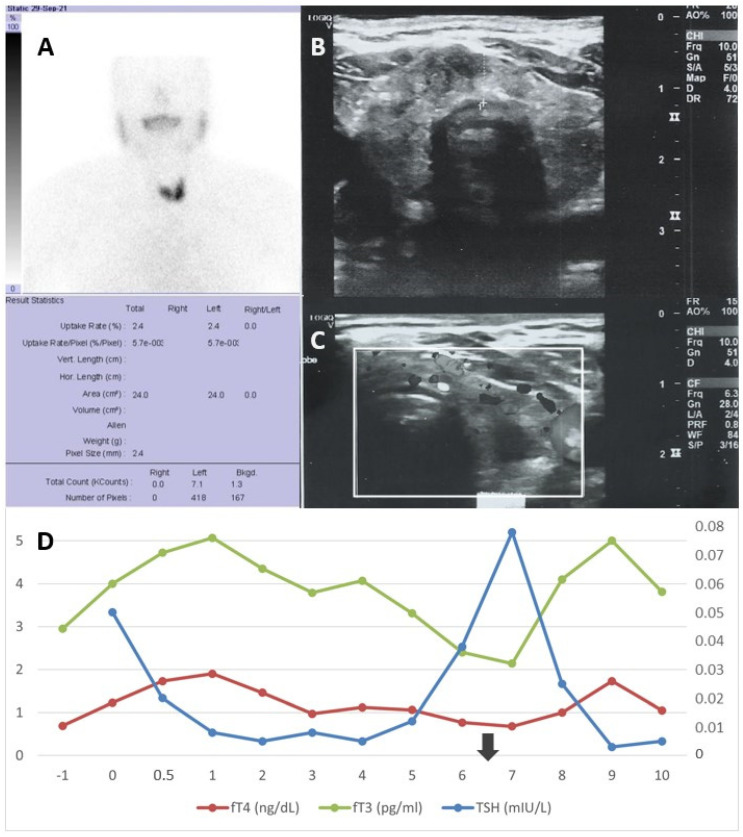
(**A**) Thyroid scintigraphy with Tc99 of case 1. The thyroid gland appears with diffuse increased activity, slightly more intense by the inferior left lobe. (**B**) Thyroid sonography of case 1. The gland appears heterogeneous, without nodules. (**C**) Thyroid Doppler sonography of case 1. Diffuse hypervascularity is observed (white frame). (**D**) Fluctuations of TSH in mIU/L, fT3 in pg/mL and fT4 in ng/dL of case 1 from baseline (three months before diagnosis) to date. Time since diagnosis is shown on *x*-axis in months. The fT3 and fT4 levels are plotted on the left *y*-axis and TSH levels are on the right one. The arrow represents time of infection.

**Figure 2 ijms-23-11492-f002:**
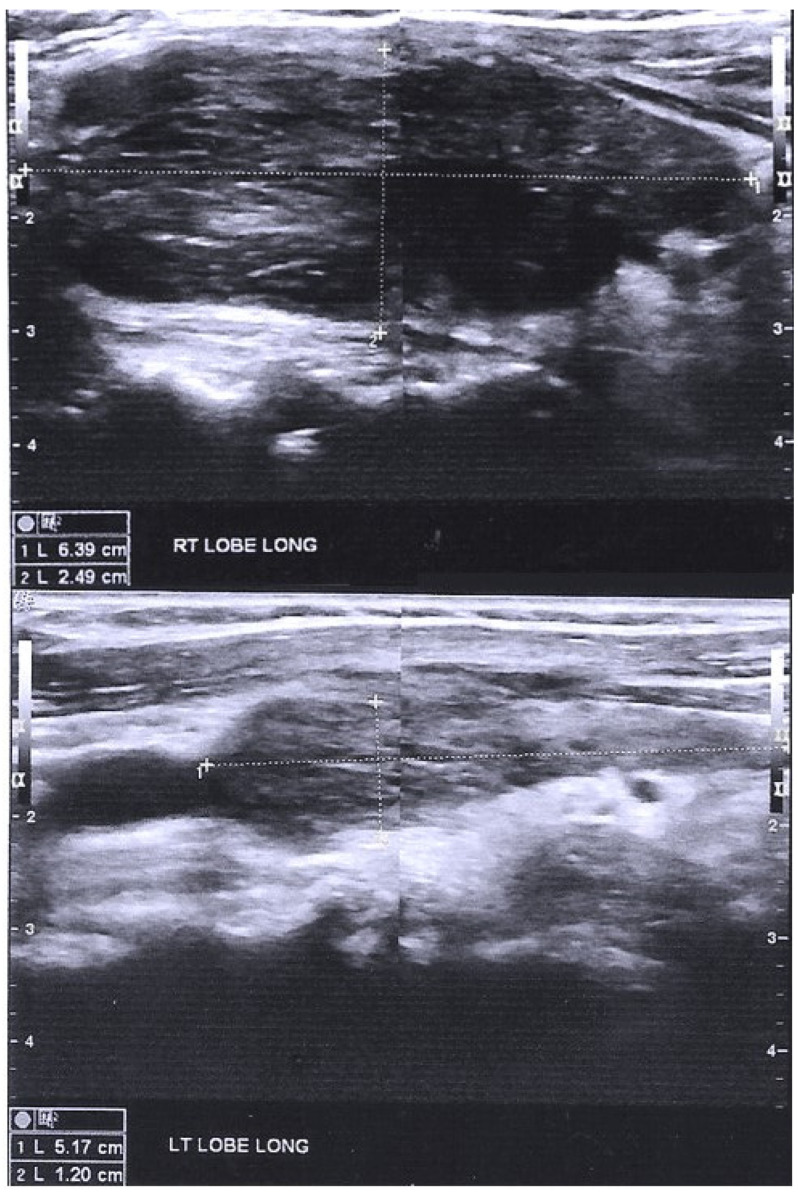
Thyroid sonography of case 2 patient. Thyroid gland is depicted enlarged (Right Lobe 6.39 × 2.49 cm, Left Lobe 5.17 × 1.20 cm) and hypoechoic, with echogenic fibrous septations. Dotted line depicts measurements of thyroid lobes.

## Data Availability

Original data are available upon request.

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
