# Peer review of "Two Cases of Autoimmune Thyroid Disorders after COVID Vaccination in Dialysis Patients"

_ijms, 2022, doi:10.3390/ijms231911492_

Round 1
Reviewer 1 Report
Review of case report entitled “Two cases of Autoimmune thyroid disorders after COVID vaccination in dialysis patients” by Georgios Lioulios et al.
The authors described two dialysis patients with no previous history of thyroid disease who developed immune mediated thyroid disorders after BNT162b mRNA vaccine against SARS-CoV-2. The first patient, a 29-year-old man on hemodialysis diagnosed with Grave’s disease four months post vaccination and the second one, a 67-year-old female on peritoneal dialysis, who developed Hashimoto thyroiditis two months post vaccination.
Although, the studies are well conducted and the clinical cases are properly described, there are some criticisms:
1. 1. The authors should report that a contributing cause in the pathogenesis of immune-based disease onset after SARS-CoV-2 vaccination could also be due to late stage ESRD-induced immunosuppression, which could play an important role in determining higher frequency of long SARS-CoV-2 infections and therefore in the insurgence of other cardiovascular and inflammatory diseases related, but also in the alterations in the immune-mediated response, that could be a role in the autoimmune diseases.
2. 2. The causal link between the onset of the two diseases and vaccination remains weak as there is no "de facto" scientific evidence to explain the pathogenetic mechanism. The authors should stress this point in the discussion, highlighting the limit of their study and giving emphasis to Graves' disease, since as they themselves report the Hashimoto thyroiditis is a frequent occurrence among patients with renal insufficiency and subject to the dialytic treatment.
3. 3. There are some spelling mistakes in the text that should be corrected.
Author Response
Thank you for your review on our paper and for your valuable comments, which helped us to improve the quality of our manuscript. We have answered each of your points below:
- The authors should report that a contributing cause in the pathogenesis of immune-based disease onset after SARS-CoV-2 vaccination could also be due to late stage ESRD-induced immunosuppression, which could play an important role in determining higher frequency of long SARS-CoV-2 infections and therefore in the insurgence of other cardiovascular and inflammatory diseases related, but also in the alterations in the immune-mediated response, that could be a role in the autoimmune diseases.
- Thank you for your useful remark. Indeed, immune alterations in ESRD are severe and have a great negative impact both in immunity against naturally occurring infections, SARS-CoV-2 included and in response to vaccination. Hence, we have added a relative paragraph in the manuscript. Please refer to lines 135-149 in the discussion section.
- The causal link between the onset of the two diseases and vaccination remains weak as there is no "de facto" scientific evidence to explain the pathogenetic mechanism. The authors should stress this point in the discussion, highlighting the limit of their study and giving emphasis to Graves' disease, since as they themselves report the Hashimoto thyroiditis is a frequent occurrence among patients with renal insufficiency and subject to the dialytic treatment.
2. Thank you very much for commenting on these issues. Arguably, there is no direct proof of a causative mechanism that links vaccination against Covid to thyroid disturbances. Indeed, while hypothyroidism is common among ESRD patients, autoimmune thyroid disturbances and especially Grave’s disease are quite rare. However, this fact is not sufficient to establish causality and thus we highlight it in the text. Please refer to lines 165-167 in the discussion
- There are some spelling mistakes in the text that should be corrected.
- Thank you for this remark. The text has been thoroughly checked for spelling and grammar oversights
Reviewer 2 Report
The manuscript entitled: “Two cases of Autoimmune thyroid disorders after COVID vaccination in dialysis patients” is a case report manuscript that presents for the first time the causality between SARS-CoV-2 vaccination and autoimmune thyroid disorders in two patients following dialysis. The manuscript is well written and very interesting for the research community and worst to be published. I have some comments for the authors to improve their manuscript before acceptance:
1. All the abbreviations should be defined at the first use in the main text, even if they were defined in the abstract, eg: ESRD, line 34, PD line 86, AL amyloidosis line 87, TSH, fT3, fT4, anti-TG, Anti-TPO
2. In figure 1B please clearly indicate the presence of the diffuse hypervascularity in the figure and improve the resolution of figure 1D
3. In figure 2 can you increase the font for the scale as in not visible and the measurements
4. Line141 please clarify what means 4 from “COVID-19infection4”, is a reference that is missing?
5. Please include some limitations of this case report
Author Response
Response to reviewer 2:
Thank you for your very useful comments and suggestions. We provide responses below, after we have corrected any indicated points.
- All the abbreviations should be defined at the first use in the main text, even if they were defined in the abstract, eg: ESRD, line 34, PD line 86, AL amyloidosis line 87, TSH, fT3, fT4, anti-TG, Anti-TPO
- Thank you for this remark. All the text has been thoroughly checked for unexplained abbreviations and corrected
- In figure 1B please clearly indicate the presence of the diffuse hypervascularity in the figure and improve the resolution of figure 1D
- Thank you for your instructive criticism. Hypervascularity is seen by a color doppler ultrasound which was not included in the original figure, due to low quality of the available image. In this version of figure 1, we include panel C, which depicts a capture of a doppler ultrasound from the same examination study, as the one seen in panel B, which unfortunately was not available in color, as this is how it was delivered to us by the ultrasound laboratory, printed on paper. Moreover, we improved resolution of all images included in figure one.
- In figure 2 can you increase the font for the scale as in not visible and the measurements
- Thank you for this comment. Unfortunately, we are not able to increase the font of the scale in figure 2, as this is the image we received printed on paper. However, we improved the image resolution and enlarged the whole figure to make it more legible.
- Line141 please clarify what means 4 from “COVID-19infection4”, is a reference that is missing?
- Thank you for this observation. The mistake was corrected (line 157).
- Please include some limitations of this case report
- Limitations of this paper were added in the discussion section. Please refer to lines 165-169
Reviewer 3 Report
No comments
accept as is
Author Response
Response to reviewer 3:
Thank you for your supporting review.
Round 2
Reviewer 1 Report
The authors have revised the text, responding adequately to the criticisms.
Reviewer 2 Report
The authors addressed all my comments. The manuscript is ready for acceptanței.